# Predicting 30-Day and 180-Day Mortality in Elderly Proximal Hip Fracture Patients: Evaluation of 4 Risk Prediction Scores at a Level I Trauma Center

**DOI:** 10.3390/diagnostics11030497

**Published:** 2021-03-11

**Authors:** Arastoo Nia, Domenik Popp, Georg Thalmann, Fabian Greiner, Natasa Jeremic, Robert Rus, Stefan Hajdu, Harald K. Widhalm

**Affiliations:** Department of Orthopedics and Traumatology, Clinical Division of Traumatology, Medical University of Vienna, Waehringer Guertel 18-20, 1090 Vienna, Austria; arastoo.nia@meduniwien.ac.at (A.N.); domenik.popp@meduniwien.ac.at (D.P.); georgthalmann@hotmail.com (G.T.); fabian.greiner@meduniwien.ac.at (F.G.); natasa.jeremic@meduniwien.ac.at (N.J.); rusrobert4@gmail.com (R.R.); stefan.hajdu@meduniwien.ac.at (S.H.)

**Keywords:** hip fracture, elderly, scoring systems, surgery, mortality, outcome, risk prediction, POSSUM, Charlson Comorbidity Index, Portsmouth-POSSUM, ACS-NSQIP^®^

## Abstract

This study evaluated the use of risk prediction models in estimating short- and mid-term mortality following proximal hip fracture in an elderly Austrian population. Data from 1101 patients who sustained a proximal hip fracture were retrospectively analyzed and applied to four models of interest: Physiological and Operative Severity Score for the enUmeration of Mortality and Morbidity (POSSUM), Charlson Comorbidity Index, Portsmouth-POSSUM and the American College of Surgeons National Surgical Quality Improvement Program (ACS-NSQIP^®^) Risk Score. The performance of these models according to the risk prediction of short- and mid-term mortality was assessed with a receiver operating characteristic curve (ROC). The median age of participants was 83 years, and 69% were women. Six point one percent of patients were deceased by 30 days and 15.2% by 180 days postoperatively. There was no significant difference between the models; the ACS-NSQIP had the largest area under the receiver operating characteristic curve for within 30-day and 180-day mortality. Age, male gender, and hemoglobin (Hb) levels at admission <12.0 g/dL were identified as significant risk factors associated with a shorter time to death at 30 and 180 days postoperative (*p* < 0.001). Among the four scores, the ACS-NSQIP score could be best-suited clinically and showed the highest discriminative performance, although it was not specifically designed for the hip fracture population.

## 1. Introduction

The crude incidence of hip fractures in Austria per 100,000 inhabitants increased in women from 493 to 642 by 2005 and in men from 192 to 280 by 2006 [1]. Emergency hip fracture surgeries tend to have a worse outcome in general when compared to elective hip surgery, even when adjusted for patient and surgical factors [2]. Elderly patients are at a higher perioperative risk and undergo a narrow range of surgical procedures. Hip fractures are associated with a poor prognosis in these patients, partly due to the high rate of postoperative complications. The mortality rates following hip fractures in elderly patients vary in the literature, from 2.8% to 12.1% for 30 days, and can range from 14% to 36% for one-year mortality, according to the literature [2,3].

There are a number of perioperative risk calculation methods developed to assist surgeons in calculating patients’ risk of perioperative mortality.

The American College of Surgeons National Surgical Improvement Program (ACS-NSQIP) Surgical Risk Calculator was developed in 2013 [4]. Several orthopedic-related articles have applied the ACS-NSQIP surgical risk calculator to identify the preoperative risk factors that increase the risk of postoperative complications in patients; however, these studies mostly focused on a single risk factor [5]. Edelstein et al. evaluated the ability of the surgical calculator to predict complications within 30 days in patients with total hip and knee arthroplasty [6] (Appendix A
Table A1).

The Physiological and Operative Severity Score for the enUmeration of Mortality and Morbidity (POSSUM) and Portsmouth-POSSUM (P-POSSUM) are general surgical tools used to efficiently assess the mortality and morbidity risks. The data suggest that these tools can be used in hip fracture patients to predict morbidity and mortality; however, it is unclear which score indicates a significant risk on a case-by-case basis [7]. It consists of a 12-factor, four-grade physiological score and a six-factor, four-grade operative severity score and includes parameters that are routinely collected throughout a patient’s hospital stay [8]. One potential concern of the POSSUM score, however, is that it has been shown to overpredict the risk of mortality and may require further calibration. To address this, the POSSUM scale was adjusted from exponential scoring to linear scoring during the development of the Portsmouth-POSSUM (P-POSSUM) score [9]. The Charlson Comorbidity Index (CCI), a system for the classification of severity that uses recorded data on secondary diagnoses, assigns a weight to morbidity, thereby generating the patient’s risk of death (Appendix A
Table A2).

This study addresses the lack of Austrian-based populations for the review of these systems and therefore aims to identify an optimal score to assess a patient’s fitness for surgery and to identify high-risk patients.

## 2. Materials and Methods

Data were retrospectively collected from a single cohort of patients aged 65 years or above who underwent surgery due to a proximal hip fracture between 1 January 2018 and 31 December 2019. Mortality for 30 days and 180 days were assessed. Patients presenting with fractures of the middle or distal third of the femur, periprosthetic femoral fractures, fractures due to polytrauma and pathological fractures were excluded (Figure 1). In the case of incomplete records with more than 10% variables missing, making it impossible to complete the used risk models, these were rejected and unable to participate in this study.

The following surgical data were collected: hip fracture type (intracapsular, extracapsular or subtrochanteric); type of surgery (total hip arthroplasty, hemiarthroplasty, triple hip screw or sliding hip screw osteosynthesis) and time to surgery (<24 h, 24 h, 24–48 h or >72 h, dating from the hospital admission to the beginning of surgery).

Mortality data in the form of month and year of death were collected from the Austrian Death Register and linked to 30-, as well as 180-day, mortality. Patients not present on the register were assumed to remain alive at the time of data collection.

As a standard procedure, the preoperative setting in our clinic was performed by a multidisciplinary medical team consisting of trauma surgeons, anesthesiologists and internal medicine specialists. Intracapsular fractures, if not displaced, were treated by internal fixation, whereas, if displaced, by arthroplasty. Extracapsular fractures such as inter- and subtrochanteric fractures were treated by internal fixation using short or long nails.

### 2.1. Ethics Approval

The study protocol conformed to the ethical guidelines of the 1975 Declaration of Helsinki, as reflected in a priori approval by the ethics board of the Medical University of Vienna. According to the committee, individual informed consent was unnecessary due to the observational characteristics of this study. No data analysis or follow-up was started prior to this study, which was approved by the institutional review and ethics board (EK No. 1517/2020) on the 25th of February 2020. Causes of death were obtained by linkage with the registry of deaths from Statistics Austria (the Austrian federal institute for Statistics). The present report was drafted in line with the STROBE statement for observational studies in epidemiology.

### 2.2. Statistical Analysis

Data were expressed as mean ± standard deviation (SD) if the data were normally distributed or as median (interquartile range (IQR)) if not. Categorial data were expressed in numbers (percentage). Normal distribution was verified using the Shapiro-Wilk’s test. Variables were compared in a Student’s *t*-test, a Wilcoxon Mann–Whitney test, a chi-square test or a Fisher’s exact test, as appropriate. Univariate analysis was used to identify risk factors of death at 6 months. Cox proportional hazards were used to display the adjusted cumulative hazard of the mortality at 6 months when *p* < 0.05 in the univariate analysis. Time-to-event analyses were performed with the use of Kaplan–Meier estimates. Statistical analyses were performed with SPSS software for Mac (v.21, IBM Corp., Armonk, NY, USA). The threshold for statistical significance was set at *p* < 0.05. The scores were analyzed in several ways. The area under the receiver operating characteristic (ROC) curve was analyzed for 30-day and 180-day mortality. A curve approaching the linear line indicated no predictive ability for the assessing system. The further from the linear line, the better the predictive ability.

A cut-off value of 0.5 was taken for each score to categorize a patient into the predicted deceased and alive groups (i.e., if a patient’s predicted mortality was found to be >50% by any of the scores, they were grouped as “more likely to die than not”).

## 3. Results

The baseline characteristics of the study population are presented in Table 1.

### 3.1. Overall Mortality Rate

Patients treated surgically for a proximal hip fracture showed a mortality rate of 6.1% at 30 days and 15.2% at 180 days, whereas most of the deceased patients included were over 80 years of age (see Table 2)

There is a significant association between the 30-day and 180-day mortalities and time to surgery. For 30-day mortality, 41 patients (61.2%) died when surgery occurred after 24 h, whereas 26 (38.8%) died when operated on within 24 h (*p* = 0.005). For 180-day mortality, 105 patients (58.7%) died when surgery occurred after 24 h, whereas 62 (41.4%) died when operated on within 24 h (*p* = 0.009). Patients who died by 30 days and 180 days exhibited lower hemoglobin (Hb) levels at admission in comparison with the survivors. The proportion of patients with less than 12.0 g/dL was higher in the nonsurvivors group (Table 3 and Table 4).

Mortality was also significantly higher in those with an ASA score of 3 at 30 days (74.3% versus 53.6% in survivors, *p* < 0.001) and at 180 days (77.6% versus 55.5% in survivors, *p* < 0.001). The proximal femoral nail and hemiarthroplasty were associated with a higher mortality at 30 days (59.7% versus 46.5% in survivors, *p* = 0.003 and 35.8% versus 31.8% in survivors, *p* = 0.005) and 180 days (61.1% versus 45.4% in survivors, *p* = 0.004 and 29.9% versus 31.5% in survivors, *p* = 0.005) (Table 3 and Table 4).

In the univariable analysis (Table 5), age, male gender, Hb level at admission <12.0 g/dL and heart failure were identified as significant risk factors associated with a shorter time to death at 30 days, whereas, at 180 days, the other variable additional to age, male gender, Hemoglobin (Hb) at admission <12.0 g/dL and heart failure associated with mortality in the univariable analysis was dyspnea (Table 6). No association between total blood loss, body mass index (BMI) and mortality was found. In addition, there was no association between mortality and diabetes.

### 3.2. Mortality Predicting Scores

ACS-NSQIP, POSSUM and P-POSSUM demonstrated areas under the curve (AUC) greater than 0.7 for 30-day mortality, indicating that they are capable predictors of mortality. The sensitivity and specificity analysis for the three predeveloped scores was completed with ROC curves for two time periods, 30 days and 180 days, graphically represented in Figure 2 and Figure 3. The ACS-NSQIP had the largest area under the ROC for mortality at both 30 days and 180 days, with areas under the curve (AUC) of 0.74 and 0.72, respectively. The CCI had its largest AUC for the 180-day mortality of 0.72. The areas under the ROC curve of all the models were not statistically significant from one another. The sensitivity and specificity analysis via logistic regression for the scores at the various time points is presented in Table 7 below. The within 30-day mortality sensitivity was moderate.

## 4. Discussion

This study, which was performed at a level I trauma center in Austria, evaluated the performance of four risk prediction models for 30-day and 180-day mortality in patients over 65 years of age undergoing hip fracture surgery. The most important finding of this study was that the ACS-NSQIP score could be best-suited clinically for the prediction of mortality due to its easy implementation. To our knowledge, this is the first study to evaluate the ACS-NSQIP for an elderly Austrian population after sustaining a proximal hip fracture.

The included patients and characteristics were similar to other studies. The average age of the patients following proximal hip fractures was 83.6 years, with a majority of ASA 2 or 3 patient statuses and a large proportion of women [10,11]. Our study subjects showed a 30-day mortality rate, which is in line with other studies [11,12].

### 4.1. Identification of Risk Factors for Mortality

Preoperative factors such as increased age, poor preinjury functional level, multiple medical comorbidities and male gender do influence postoperative complications and mortality and, therefore, seem to play a significant role in increasing early mortality rates [3,13].

The finding of an increased mortality risk for men with a hip fracture is consistent with several other studies [12,14]. In order to explain the gender difference in mortality after hip fracture, a study by Wehren et al. proposed that men’s health was more unstable at the time of fracture, making them vulnerable to various kinds of infections [15]. Panula et al. suggested that one reason for mortality in men could be due to respiratory problems after hip fracture surgery [16].

In the univariable analysis (Table 4 and Table 5), age, male gender, and hemoglobin (Hb) levels at admission <12.0 g/dL were identified as significant risk factors associated with a shorter time to death at 30 and 180 days; similar results were also described by Yombi et al. [14].

Furthermore, anemia was an independent associated risk factor for mortality in this study, which is comparable with the study of Kovar et al., who found an association between short-term mortality and Hb levels at admission in a large study of 3595 patients [17]. In line with the literature, it was reported that there are strong associations between mortality and the occurrence of cardiovascular diseases [18,19] on the one hand and the ASA classification on the other hand. Therefore, all patients with hip fractures underwent a preoperative assessment after admission and were graded according to ASA grading. In general, a high ASA score indicates an already significant preoperative morbidity and the need for appropriate early treatment of these patients. Based on these findings, the results of the present study showed that higher ASA grades are associated with an increase in 30-day and 180-day mortality.

Within the patient cohort of this study, a delay in surgery did raise the risk of mortality, indicating that accelerated surgery is crucial for survival. In 2014, the Hip Fracture Accelerated Surgical Treatment and Care Track (HIP ATTACK) investigators already showed that patients who were treated in the first six hours after injury (accelerated care group) had significantly lower 30-day mortality rates compared to the standard care group (3% and 13%, respectively). The study design included, on the one hand, the accelerated care group with a mean time of surgery of six hours in contrast to the standard care group of 24.2 h [20]. Nyholm et al. [21] showed that even a delay of surgery for over 12 h increased the 30-day mortality rate by 30%. In a meta-analysis by Klestil et al., 28 prospective studies with datasets from 31,242 patients were analyzed, which showed that patients who were operated on within 48 h had a 20% lower risk of death within 12 months. However, no statistically significant changes of the mortality rates could be shown when comparing patients who were operated on within or after 24 h [22].

### 4.2. Performance of Scores

Two of the four scoring systems resulted in acceptable discrimination when applied to this study population, while none of the models showed excellent discriminative powers (AUC > 0.80). ACS-NSQIP yielded the highest sensitivity and specificity, with an area under the ROC curve of 0.74 for 30-day and 0.72 for 180-day mortality, although this universal risk calculator lacks validity in arthroplasty and pulmonary, as well as orthopedic, surgery [6,22,23]. Edelstein et al. showed poor 30-day mortality performances of the ACS-NSQIP, but it has to be noted that the study cohort was elective total hip and total knee arthroplasty, and therefore, there were probably less patients with acute medical conditions. To our knowledge, there is currently no long-term analysis of the ACS-NSQIP for acute hip surgeries.

This performance was not significantly different from the CCI. The two models with the lowest discriminative power, the POSSUM and the P-POSSUM, did not demonstrate a significant lack of fit. In detail, the CCI showed a ROC of 0.70 for the 30-day and 0.72 for the 180-day mortality rate, indicating a moderate prediction power. The 30-day performance was, on the one hand, in contrast to the work of Pei-Ling et al. (AUC 0.65) [24], as well as Quach et al. [25] (AUC 0.6) but, on the other hand, similar to the findings of Karres et al. (AUC 0.71). Better results were seen in the work of Nelson et al., which showed satisfying results (AUC 0.77) with the age-adjusted CCI for 30-day mortality. A similar performance for 180-day mortality of the CCI was found by Boddaert et al. [26] (AUC 0.64), whereas Hautamäki found better results (AUC 0.77) for 180-day mortality [27]. An age-adjusted developed score might be more suitable, but further studies are needed to verify its prediction power [28].

Compared to the literature, the POSSUM, as well as the P-POSSUM, performed poorly in general, being similar to the 30-day mortality findings of Ramanathan et al. (AUC 0.62) in particular when compared to the AUC values for POSSUM in general surgery of >0.9 and general orthopedics of >0.85 [29,30]. Burgos et al. [31], Nelson et al. [28] and Karres et al. [32] also evaluated different scoring systems in hip fractures in a frail population, including the CCI and POSSUM. The authors came to different values for the CCI or POSSUM, showing a big range in the results for 30-day mortality, whereas Karres et al. [32] reported an AUC for the CCI of 0.69 and 0.71 for the POSSUM, Burgos et al. [31] had an AUC of 0.64 for the CCI and 0.59 for POSSUM and Nelson et al. had an AUC of 0.77 for CCI and 0.76 for POSSUM for 30 day-mortality. Nelson et al. also reported a better AUC of 0.74 for the POSSUM and a better AUC of 0.75 for the CCI for 180-day mortality [28].

Recently, there has been progress in the field of machine learning. Li et al. applied a random survival forest (RSF) algorithm to patients with acute hip fracture and were able to show that variables such as complications, length of stay, age and ventilation are significant for 30-day predictions and one-year mortality [33]. Machine learning in general is a promising approach and is already used in the field of image analyzing, such as for osteoarthritis, and without a doubt, research is accelerating in this area. Nevertheless, these tools must fully align with the data protection requirements and minimize any effects of bias, and the question of transparency has yet to be answered [34].

The ACS-NSQIP showed the best AUC in this study, although literature describing its ability for mortality predictions is rare. How these scores are used in daily clinical settings is a subject of discussion. These scores can help anesthesiologists and surgeons to identify high-risk patients who need special intraoperative and perioperative management.

This easy, clinically applicable scoring system could be used more systematically to tailor pre- and postoperative care. Furthermore, this tool might guide the choice of surgical treatment for patients with an acute proximal hip fracture. Based on this fact, that application of such scores is commonly used for the prediction of serious complications after general surgery; further studies in the field of orthopedic trauma are needed to prove the results of this study.

Finally, none of the evaluated models used in this study showed acceptable discrimination, and none of them achieved an AUC over 0.80, making improvements in predicting mortality after hip fracture surgery inevitable. Since these models did not show excellent discrimination, further research is needed to determine a better risk model for predicting mortality following hip fracture surgery.

### 4.3. Limitation

The major strength of this study was the analysis of a large amount of data of patients with all types of proximal hip fractures. The limitations of our study largely revolved around its retrospective design; therefore, not all necessary data was available. Intraoperative blood loss was poorly recorded, resulting in the use of an estimate of this variable. Unfortunately, the data relating to the postoperative recovery period was lacking. This study was limited to a single center; to address these concerns, a prospective study would be beneficial to further compare the available scores in an Austrian population.

## 5. Conclusions

Predicting short- and long-time mortality after the surgical treatment of elderly proximal hip fracture patients using different prediction scores is possible. In this cohort, the ACS-NSQIP scores showed the best fit rate of all four tested scores, although it was not primarily intended for use in the hip surgery field. For the first time, this study included results that exceed 30-day mortality using these scores.

## Figures and Tables

**Figure 1 diagnostics-11-00497-f001:**
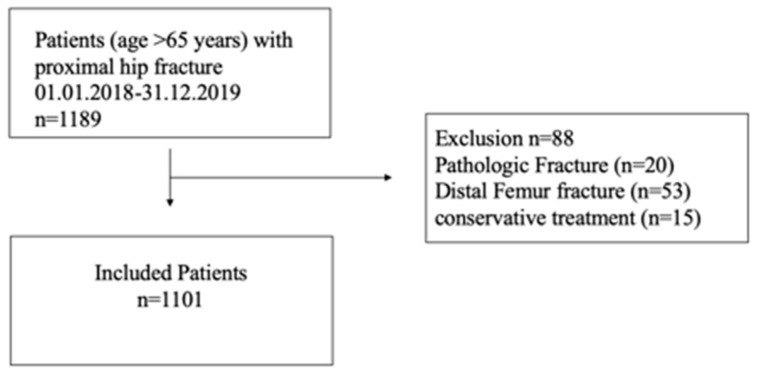
Flowchart indicating patient inclusion.

**Figure 2 diagnostics-11-00497-f002:**
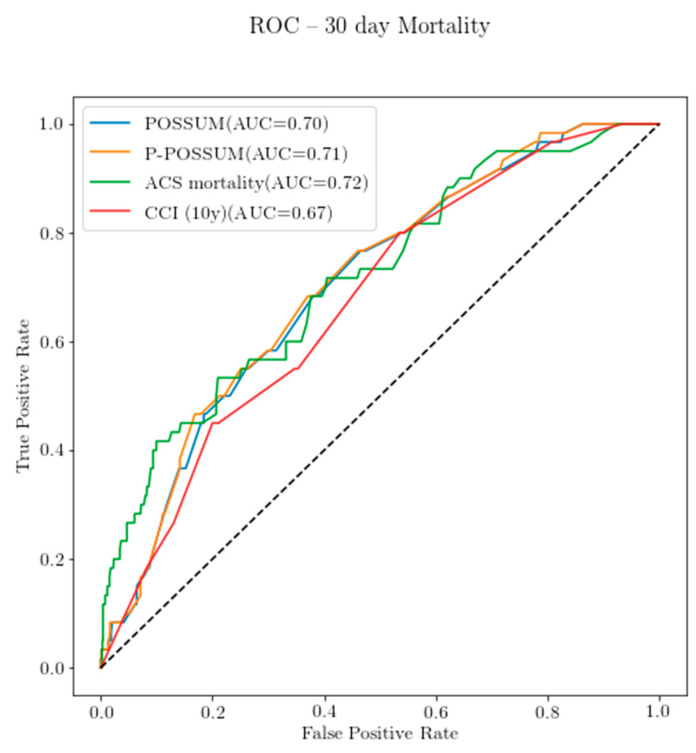
Receiver operating characteristic (ROC) curve for the 30-day mortality prediction on the calibrated dataset (*n* = 1101). POSSUM: Physiological and Operative Severity Score for the enUmeration of Mortality and Morbidity, P-POSSUM: Portsmouth-POSSUM, CCI: Charlson Comorbidity Index and ACS: American College of Surgeons.

**Figure 3 diagnostics-11-00497-f003:**
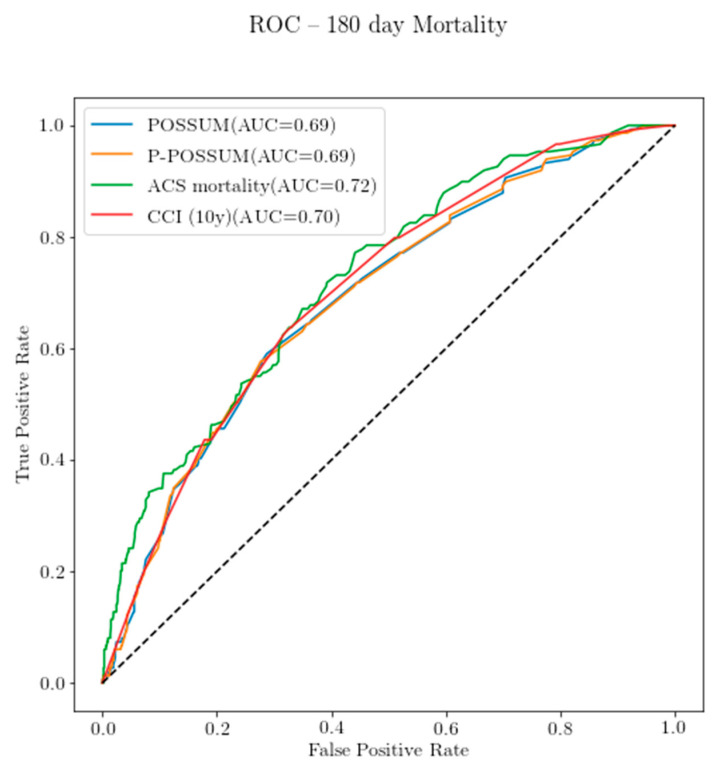
ROC curve for the 180-day mortality prediction on the calibrated dataset (*n* = 1101).

**Table 1 diagnostics-11-00497-t001:** Demographics. BMI: body mass index.

Demographics	*n*	%
total patients (*n*)	1101	
male	342	30.4
female	759	69.6
age (mean–standard deviation)	83.6 ± 9.2	
BMI (kg/m^2^)	23.81	
Comorbidities		
arterial hypertension	774	70.3
chronic obstructive disease/asthma	127	11.5
chronic heart arrythmia	175	15.9
ischemic heart disease	139	12.6
acute renal insufficiency	59	5.4
diabetes	226	20.5
Hemoglobin g/dL (mean)	11.3	
Fracture type		
Intracapsular	535	48.6
Extracapsular	556	50.5
Subtrochanteric	10	0.9
Mobility		
independent	516	46.9
partial dependent	357	32.4
dependent	228	20.7

**Table 2 diagnostics-11-00497-t002:** Mortality after hip surgery.

Mortality	(*n* = 1101)	%
Overall mortality, *n* (%)		
30 days	67	6.1
180 days	167	15.2
30-day mortality classified by age, *n* (%)		
65–70	3	0.3
70–80	16	1.5
>80	48	4.4
30-day mortality classified by sex, *n* (%)		
male	36	3.3
female	31	2.8
180-day mortality classified by age, *n* (%)		
65–70	12	1.1
70–80	33	3.0
>80	122	11.1
180-day mortality classified by sex, *n* (%)		
male	81	7.4
female	86	7.8

**Table 3 diagnostics-11-00497-t003:** Characteristics and univariate analysis of the surgical and clinical factors influencing the 30-day mortality after hip fracture surgery. Data are expressed as numbers (proportions).

Clinical Data *n* (%)			
Days until surgery	Alive *n* = 867	30 days *n* = 67	*p*-value
<24 h	394 (45.4)	11 (16.4)	
24 h	243 (28)	15 (22.4)	
24–48	129 (14.9)	19 (28.4)	0.005
>72 h	101 (11.6)	22 (32.8)	
ASA score			
1	24 (2.8)	0	
2	344 (39.7)	12 (17.9)	
3	481 (55.5)	52 (77.6)	0.001
4	18 (2.1)	3 (4.5)	
Hemoglobin g/dL (mean)	12.2	10.7	0.001
Type of surgery			
Hip hemiarthroplasty	276 (31.8)	24 (35.8)	0.005
Total hip arthroplasty	62 (7.2)	1 (1.5)	
Proximal femoral nail	403 (46.5)	40 (59.7)	0.003
Double screw method	70 (8.1)	2 (3.0)	
Sliding hip screw	56 (6.5)	0	

**Table 4 diagnostics-11-00497-t004:** Characteristics and univariate analysis of the surgical and clinical factors influencing 180-day mortality after hip fracture surgery. Data are expressed as numbers (proportions).

Clinical Data *n* (%)			
Days until surgery	Alive *n* = 800	180 days *n* = 167	*p*-value
<24 h	372 (46.5)	32 (23.4)	
24 h	228 (28.5)	30 (18)	
24–48	110 (13.75)	66 (39.5)	0.009
>72 h	90 (11.25)	39 (19.2)	
ASA score			
1	24 (3)	0	
2	332 (41.5)	33 (19.8)	
3	429 (53.6)	124 (74.3)	0.001
4	15 (1.9)	10 (6.0)	
Hemoglobin g/dL (mean)	12.2	11.1	0.001
Type of surgery			
Hip hemiarthroplasty	252 (31.5)	50 (29.9)	0.005
Total hip arthroplasty	61 (7.6)	3 (1.8)	
Proximal femoral nail	363 (45.4)	102 (61.1)	0.004
Double screw method	68 (8.5)	11 (6.6)	
Sliding hip screw	56 (7)	1 (0.6)	

**Table 5 diagnostics-11-00497-t005:** Univariable Cox regression model predicting the hazard ratio for 30-day mortality.

Covariate	Coefficient	Coef. Lower 95%	Coef. Upper 95%	*p*-Value
Total Blood Loss (mL)	0.00	0.00	0.00	0.22
Age	0.07	0.04	0.10	<0.001 *
Hemoglobin (g/dL)	−0.31	−0.42	−0.20	<0.001 *
BMI	0.01	−0.04	0.07	0.64
Smoking	0.05	−0.74	0.84	0.90
Diabetes	1.58	−0.39	3.56	0.12
Arterial Hypertension	0.69	0.04	1.35	0.04
Ischemic cardiopathy	0.47	−0.16	1.11	0.14
Heart Failure	1.09	0.51	1.68	<0.001 *
Dyspnea	0.80	0.17	1.43	0.01
COPD/asthma	0.34	−0.37	1.05	0.35
Gender Male	0.95	0.45	1.46	<0.001 *
Statistically significant	*			

**Table 6 diagnostics-11-00497-t006:** Univariable Cox regression model predicting the hazard ratio for 180-day mortality.

Covariate	Coef	Coef Lower 95%	Coef Upper 95%	*p*-Value
Total Blood Loss (mL)	0.00	0.00	0.00	0.90
Age	0.06	0.04	0.07	<0.001 *
Hemoglobin (g/dL)	−0.26	−0.33	−0.19	<0.001 *
BMI	−0.02	−0.06	0.02	0.35
Smoking	−0.04	−0.56	0.48	0.89
Diabetes	0.74	−1.23	2.70	0.46
Arterial Hypertension	0.14	−0.22	0.50	0.44
Ischemic cardiopathy	0.33	−0.10	0.75	0.13
Heart Failure	1.00	0.61	1.39	<0.001 *
Dyspnea	0.77	0.35	1.18	<0.001 *
COPD/asthma	0.60	0.18	1.01	0.01
Gender Male	0.86	0.54	1.18	<0.001 *
Statistically significant	*			

**Table 7 diagnostics-11-00497-t007:** AUC of risk prediction scores for 30-day and 180-day mortality. POSSUM: Physiological and Operative Severity Score for the enUmeration of Mortality and Morbidity, P-POSSUM: Portsmouth-POSSUM, CCI: Charlson Comorbidity Index and ACS: American College of Surgeons.

Score	30-Day	180-Day
POSSUM	0.70	0.69
P-POSSUM	0.71	0.69
CCI	0.67	0.70
ACS Mortality	0.72	0.72

## Data Availability

The datasets generated during and/or analyzed during the current study are not publicly available due to data privacy but are available from the corresponding author upon reasonable request. There is no public access to the hospital patient data used due to data privacy. Administrative permission was given by the local ethics committee, the Medical University of Vienna, as it represents a standard procedure for any study performed at the Medical University of Vienna.

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
