# Peer review of "Predicting 30-Day and 180-Day Mortality in Elderly Proximal Hip Fracture Patients: Evaluation of 4 Risk Prediction Scores at a Level I Trauma Center"

_diagnostics, 2021, doi:10.3390/diagnostics11030497_

Round 1

Reviewer 1 Report

Study well done and interesting. However, despite the conclusins in the study, in a patient identified at risk, there is not much time to improve his conditions.

Author Response

Comment: Study well done and interesting. However, despite the conclusins in the study, in a patient identified at risk, there is not much time to improve his conditions.

Response: Thank you for pointing this out. We agree with your comment, since several studies showed that after all surgeries within 6 hours after admission have the lowest mortality rate and even the controversial preoperative echocardiography showed no better results. Nevertheless, these scores, that need just a couple of minutes,  can help trauma staff by raising awareness for patients with high risk of post surgery complications. E.g. the ACS-NSQIP shows in particular a summary of each possible complication and therefore could help especially trauma teams in tertiary hospitals , where its not common to have a geriatrician involved.

Reviewer 2 Report

Predicting 30-day and 180-day mortality in elderly proximal hip fracture patients: Evaluation of 4 risk prediction scores at a level I trauma center   This study evaluated the use of risk prediction models in estimating short- and mid-term mortality following proximal hip fracture in an elderly Austrian population. Data from 1101 patients who have sustained a proximal hip fracture were retrospectively analyzed and applied to four models of interest: POSSUM, Charlson Comorbidity Index, Portsmouth-POSSUM and the American College of Surgeons National Surgical Quality Improvement Program (ACS NSQIP®) Risk Score. The performance of these models according to risk prediction of short- and mid-term mortality was assessed with receiver operating characteristic curve (ROC).  Age, male gender, and Hemoglobin (Hb) levels at admission < 12.0 g/dl, were identified as significant risk factors associated with shorter time to death at 30- and 180-days postoperative (p < 0.001). Among 4 scores, the ACS-NSQIP score could be best suited clinically and showed the highest discriminative performance although it was not specifically designed for the hip fracture population.   The study was well designed and implemented. No major issues, but for the general readers it is mandatory to add a few further REFERENCES, briefly discussing them. A few examples follow:   Ricciardi C , Jónsson H Jr., Jacob D, Improta G, Recenti M, Gíslason MK, Cesarelli G , Esposito L, Minutolo V, Bifulco P and Gargiulo P. Improving Prosthetic Selection and Predicting BMD from Biometric Measurements in Patients Receiving Total Hip Arthroplasty. Diagnostics 2020, 10, 0815; doi:10.3390/diagnostics10100815   Recenti, M.; Ricciardi, C.; Edmunds, K.; Gislason, M.K.; Gargiulo, P. Machine learning predictive system based upon radiodensitometric distributions from mid-thigh CT images. Eur. J. Transl. Myol. 2020, 30, 8892, doi:10.4081/ejtm.2019.8892.   Edmunds, K.J.; Gíslason, M.K.; Arnadottir, I.D.; Marcante, A.; Piccione, F.; Gargiulo, P. Quantitative Computed Tomography and Image Analysis for Advanced Muscle Assessment. Eur. J. Transl. Myol. 2016, 26, 6015, doi:10.4081/ejtm.2016.6015.   Gargiulo, P.; Gislason, M.K.; Edmunds, K.J.; Pitocchi, J.; Carraro, U.; Esposito, L.; Fraldi, M.; Bifulco, P.; Cesarelli, M.; Jónsson, H. CT-Based Bone and Muscle Assessment in Normal and Pathological Conditions. Encycl. Biomed. Eng. 2019, 3, 119–134. [CrossRef]
Gislason, M.K.; Lupidio, F.; Jónsson, H.; Cristofolini, L.; Esposito, L.; Bifulco, P.; Fraldi, M.; Gargiulo, P. Three dimensional bone mineral density changes in the femur over 1 year in primary total hip arthroplasty patients. Clin. Biomech. 2020, 78, 105092. [CrossRef] [PubMed]   Pétursson, Þ.; Edmunds, K.J.; Gislason, M.K.; Magnússon, B.; Magnúsdóttir, G.; Halldórsson, G.; Jónsson, H.; Gargiulo, P. Bone Mineral Density and Fracture Risk Assessment to Optimize Prosthesis Selection in Total
Hip Replacement. Comput. Math. Methods Med. 2015, 2015, 1–7.

Author Response

Comment 1: The study was well designed and implemented. No major issues, but for the general readers it is mandatory to add a few further REFERENCES, briefly discussing them. A few examples follow:   Ricciardi C , Jónsson H Jr., Jacob D, Improta G, Recenti M, Gíslason MK, Cesarelli G , Esposito L, Minutolo V, Bifulco P and Gargiulo P. Improving Prosthetic Selection and Predicting BMD from Biometric Measurements in Patients Receiving Total Hip Arthroplasty. Diagnostics 2020, 10, 0815; doi:10.3390/diagnostics10100815   Recenti, M.; Ricciardi, C.; Edmunds, K.; Gislason, M.K.; Gargiulo, P. Machine learning predictive system based upon radiodensitometric distributions from mid-thigh CT images. Eur. J. Transl. Myol. 2020, 30, 8892, doi:10.4081/ejtm.2019.8892.   Edmunds, K.J.; Gíslason, M.K.; Arnadottir, I.D.; Marcante, A.; Piccione, F.; Gargiulo, P. Quantitative Computed Tomography and Image Analysis for Advanced Muscle Assessment. Eur. J. Transl. Myol. 2016, 26, 6015, doi:10.4081/ejtm.2016.6015.   Gargiulo, P.; Gislason, M.K.; Edmunds, K.J.; Pitocchi, J.; Carraro, U.; Esposito, L.; Fraldi, M.; Bifulco, P.; Cesarelli, M.; Jónsson, H. CT-Based Bone and Muscle Assessment in Normal and Pathological Conditions. Encycl. Biomed. Eng. 2019, 3, 119–134. [CrossRef] 
Gislason, M.K.; Lupidio, F.; Jónsson, H.; Cristofolini, L.; Esposito, L.; Bifulco, P.; Fraldi, M.; Gargiulo, P. Three dimensional bone mineral density changes in the femur over 1 year in primary total hip arthroplasty patients. Clin. Biomech. 2020, 78, 105092. [CrossRef] [PubMed]   Pétursson, Þ.; Edmunds, K.J.; Gislason, M.K.; Magnússon, B.; Magnúsdóttir, G.; Halldórsson, G.; Jónsson, H.; Gargiulo, P. Bone Mineral Density and Fracture Risk Assessment to Optimize Prosthesis Selection in Total
Hip Replacement. Comput. Math. Methods Med. 2015, 2015, 1–7.  

Response: Thank you for this suggestions. We agree with this and have incorporated your suggestion in the manuscript, nevertheless it would have been interesting to explore the whole extent of this aspect. However, in the case of our study, we are testing current common scores used in the daily clinical practice, therefore this would go beyond the initial scope of our work. We have highlighted the changes within the manuscript.
